# Recognising and evaluating the effectiveness of extortion in the Iterated Prisoner's Dilemma

Vincent Knight[1]*, Marc Harper[2], Nikoleta E. Glynatsi[3], Jonathan Gillard[1]

**1** School of Mathematics, Cardiff University, Cardiff, United Kingdom, **2** Google Inc., Mountain View, CA, United States of America, **3** Max Planck Research Group on the Dynamics of Social Behavior, Max Planck Institute for Evolutionary Biology, Plön, Germany

☙ These authors contributed equally to this work.
* knightva@cardiff.ac.uk

**Data Availability Statement:** All of the code and data discussed in Results section is 50 open sourced, archived, and written according to best scientific principles. The data archive can be found at https://doi.org/10.5281/zenodo.1297075. and

## Abstract

Establishing and maintaining mutual cooperation in agent-to-agent interactions can be viewed as a question of direct reciprocity and readily applied to the Iterated Prisoner's Dilemma. Agents cooperate, at a small cost to themselves, in the hope of obtaining a future benefit. Zero-determinant strategies, introduced in 2012, have a subclass of strategies that are provably extortionate. In the established literature, most of the studies of the effectiveness or lack thereof, of zero-determinant strategies is done by placing some zero-determinant strategy in a specific scenario (collection of agents) and evaluating its performance either numerically or theoretically. Extortionate strategies are algebraically rigid and memory-one by definition, and requires complete knowledge of a strategy (the memory-one cooperation probabilities). The contribution of this work is a method to detect extortionate behaviour from the history of play of an arbitrary strategy. This inverts the paradigm of most studies: instead of observing the effectiveness of some theoretically extortionate strategies, the largest known collection of strategies will be observed and their intensity of extortion quantified empirically. Moreover, we show that the lack of adaptability of extortionate strategies extends via this broader definition.

## Introduction

The Iterated Prisoner's Dilemma (IPD) is a model for rational and evolutionary interactive behaviour, having applications in biology, the study of human social behaviour, and many other domains. A standard representation of the game is given in Eq 1, where the constraints ensure a non-cooperative equilibrium.

$$\begin{pmatrix} R & S \\ T & P \end{pmatrix} \qquad T > R > P > S \text{ and } 2R > T + S \qquad (1)$$

The parameters of (1) correspond to:

the source code was developed at https://github.com/drvinceknight/testing_for_ZD/ and has been archived at https://doi.org/10.5281/zenodo.2598534.

**Funding:** The author(s) received no specific funding for this work.

**Competing interests:** NO authors have competing interests.

- *R*: The reward for both players cooperating.

- *T*: The temptation value of defecting.

- *S*: The sucker value of cooperating against a defection.

- *P*: The punishment value when both players defect.

Early work in the field [1, 2] showed that cooperative behaviour could be successful in repeated interactions: Tit For Tat performed strongly in a tournament of strategies with various degrees of non-cooperation. The simplicity of Tit For Tat, which only requires knowledge of the opponent's previous play, led to much research concentrating on these so called memory-one strategies. A bibliometric study of the literature on the IPD is available at [3].

A subclass of memory-one strategies known as zero-determinant (ZD) strategies were introduced in [4]. Of these, extortionate strategies have received considerable interest in the literature [5]. These strategies "enforce" a difference in stationary payouts between themselves and their opponents. The definition requires a precise algebraic relationship between the probabilities of cooperation given the outcome of the previous round of play. Slight alterations to these probabilities can cause a strategy to no longer satisfy the necessary relations to be considered extortionate.

In [5–11] the effectiveness of these strategies in an evolutionary setting was discussed. For example [6] showed that ZD strategies were not evolutionarily stable. Furthermore, in that work it was also postulated that 'evolutionarily successful ZD strategies could be designed that use longer memory to distinguish self from non-self'. In [12] long memory strategies are designed that are able to self recognise and in [11] evolutionary processes showed the emergence of similar abilities. In [7] two sets of strategies are identified: partners and rivals and some discussion about the environments necessary for either to be evolutionary stable are given. In a non-evolutionary context, the work of [13] uses social experiments to suggest that higher rewards promote extortionate behaviour, where statistical techniques are used to identify such behaviour.

The algebraic relationships of extortion, discussed in the Methods section, define a subspace of $\mathbf{p} \in \mathbb{R}^4$ which can be used to broaden the definition of an extortionate strategy by requiring only that the defining four cooperation probabilities of a memory-one strategy are close to an algebraically extortionate strategy, by the usual technique of orthogonal projection. Moreover, given the history of play of a strategy in an actual matchup, we can empirically observe its four cooperation probabilities, measure the distance to the subspace of extortionate strategies, and use this distance as a measure of the extortionality of a strategy. This method can be applied to any strategy regardless of the memory depth and avoids the algebraic rigidity and instability issues.

We apply this method to the largest known corpus of strategies for the iterated prisoner's dilemma (the Axelrod Python library [14, 15]) and validate empirically that the method in fact detects extortionate strategies. A large tournament with 204 strategies demonstrates that sophisticated strategies can in fact recognise extortionate behaviour and adapt to their opponents. Further, statistical analysis of these strategies in the context of evolutionary dynamics demonstrates the importance of adaptability to achieve evolutionary stability. All of the code and data discussed in Results section is open sourced, archived, and written according to best scientific principles [16]. The data archive can be found at [17] and the source code was developed at https://github.com/drvinceknight/testing_for_ZD/ and has been archived at [18]. This large tournament is complemented with evolutionary dynamics that offer some insight into the effectiveness of extortionate strategies.

Several theoretical insights emerge from this work. Infamously, extortionate strategies do not play well with themselves. In [4], Press and Dyson claim that a player with a "theory of

mind" would rationally chose to cooperate against an opponent that also has knowledge of zero-determinant strategies to avoid sustained mutual defection. While not possible for memory-one strategies, we show that this behavior is exhibited by relatively simple longer memory strategies which previously emerged from an evolutionary selection process. Similarly, in [6], Adami and Hintze suggest that there may exist strategies that are able to selectively behave extortionately to some opponents and cooperatively to others. We show that this is indeed the case for the same evolved strategies. It seems that humans have trouble explicitly creating such strategies but evolution is able to do so by optimizing for total payoff in IPD interactions. Accordingly, while resistance to extortionate behavior appears critical to the evolution of cooperation, there is no prohibition on selectively extorting weaker opponents, even in population dynamics, and this behavior is evolutionarily advantageous.

## Materials and methods

### Recognising extortion

This section reviews the definition of ZD strategies from the literature, present the vector space in which such strategies exist and finally present a novel measure that allows for a measure of how far any memory-one strategy is from the space of ZD strategies. Note that in this section no claims about the evolutionarily effectiveness of such strategies are made.

ZD strategies are a special case of memory-one strategies, which are defined by elements of $\mathbb{R}^4$, mapping a state of $\{C, D\}^2$, corresponding to the prior round of play, to a probability of cooperating in the next round. A match between two such strategies creates a Markov chain with transient states $\{C, D\}^2$. The main result of [4] is that given two memory-one players $\mathbf{p}, \mathbf{q} \in \mathbb{R}^4$, a linear relationship between the players' scores can, in some cases, be forced by one of the players for specific choices of these probabilities.

Using the notation of [4], the utilities for player $X$ (playing the strategy $\mathbf{p}$) are given by $\mathbf{S}_x = (R, S, T, P)$ and for player $Y$ (playing the strategy $\mathbf{q}$ by $\mathbf{S}_y = (R, T, S, P)$ and the stationary scores of each player are given by $S_X$ and $S_Y$ respectively. The main result of [4] is that if

$$\tilde{\mathbf{p}} = \alpha \mathbf{S}_x + \beta \mathbf{S}_y + \gamma \tag{2}$$

or

$$\tilde{\mathbf{q}} = \alpha \mathbf{S}_x + \beta \mathbf{S}_y + \gamma \tag{3}$$

where $\tilde{\mathbf{p}} = (p_1 - 1, p_2 - 1, p_3, p_4)$ and $\tilde{\mathbf{q}} = (q_1 - 1, q_3, q_2 - 1, q_4)$ then:

$$\alpha S_X + \beta S_Y + \gamma = 0 \tag{4}$$

Extortionate strategies are defined as follows. If this relationship is satisfied

$$\gamma = -P(\alpha + \beta) \tag{5}$$

then the player can ensure $(S_X - P) = \chi(S_Y - P)$ where:

$$\chi = \frac{-\beta}{\alpha} \tag{6}$$

Thus, if Eq (5) holds and $\chi > 1$ then a player is said to extort their opponent. Next, the reverse problem is considered: given a $\mathbf{p} \in \mathbb{R}^4$ can one determine if the associated strategy is attempting to act in an extortionate way?

## Subspace of extortionate strategies

Constraints (2) and (5) correspond to:

$$\tilde{p}_1 = \alpha R + \beta R - P(\alpha + \beta) \tag{7}$$

$$\tilde{p}_2 = \alpha S + \beta T - P(\alpha + \beta) \tag{8}$$

$$\tilde{p}_3 = \alpha T + \beta S - P(\alpha + \beta) \tag{9}$$

$$\tilde{p}_4 = \alpha P + \beta P - P(\alpha + \beta) = 0 \tag{10}$$

Eq (10) ensures that $p_4 = \tilde{p}_4 = 0$. Eqs (7)–(9) can be used to eliminate $\alpha, \beta$, giving:

$$\tilde{p}_1 = \frac{(R - P)(\tilde{p}_2 + \tilde{p}_3)}{S + T - 2P} \tag{11}$$

with:

$$\chi = \frac{\tilde{p}_2(P - T) + \tilde{p}_3(S - P)}{\tilde{p}_2(P - S) + \tilde{p}_3(T - P)} \tag{12}$$

Given a strategy $p \in \mathbb{R}^4$ Eqs (10)–(12) can be used to check if a strategy is extortionate. The conditions correspond to:

$$p_1 = \frac{(R - P)(p_2 + p_3) - R + T + S - P}{S + T - 2P} \tag{13}$$

$$p_4 = 0 \tag{14}$$

$$p_2 + p_3 < 1 \tag{15}$$

The algebraic steps necessary to prove these results are available in the supporting materials, and note that an equivalent formulation was obtained in [6].

Based on Eqs (13)–(15), it is evident that all extortionate strategies reside on a triangular plane within a three-dimensional space. Using this formulation it can be seen that a necessary (but not sufficient) condition for an extortionate strategy is that it cooperates on average less than 50% of the time when in a state of disagreement with the opponent (15).

As an example, consider the known extortionate strategy **p** = (8/9, 1/2, 1/3, 0) from [19], which is referred to as Extort-2. In this case, for the standard values of $(R, S, T, P) = (3, 0, 5, 1)$ constraint (13) corresponds to:

$$p_1 = \frac{2(p_2 + p_3) + 1}{3} = \frac{2(1/2 + 1/3) + 1}{3} = \frac{8}{9} \tag{16}$$

It is clear that in this case all constraints hold. As a counterexample, consider the strategy that cooperates 25% of the time: **p** = (1/4, 1/4, 1/4, 1/4) satisfies (15) but is not extortionate as:

$$p_1 \neq \frac{2(p_2 + p_3) + 1}{3} = \frac{2(1/4 + 1/4) + 1}{3} = \frac{2}{3} \tag{17}$$

## Measuring extortion from the history of play

Not all strategies are memory-one strategies but it is possible to measure a given **p** from any set of interactions between two strategies. This approach can then be used to confirm that a given strategy is acting in an extortionate manner even if it is not a memory-one strategy. However, in practice, if an exact form for **p** is not known but measured from observed plays of the game then measurement and/or numerical error might lead to an extortionate strategy not being confirmed as such. Comparing theoretic and actual plays of the IPD is not novel, see for example [20].

As an example consider Table 1, which shows some actual plays of Extort-2 (**p** = (8/9, 1/2, 1/3, 0)) against an alternating strategy (**p** = (0, 0, 1, 1)). In this particular instance the measured value of **p** for the known extortionate strategy would be: (2/2, 1/5, 3/8, 0/4) which does not fit the definition of a ZD strategy.

Note that measurement of behaviour might in some cases lead to missing values. For example the strategy **p** = (8/9, 1/2, 1/3, 0) when playing against an opponent that always cooperates will in fact never visit any state which would allow measurement of $p_3$ and $p_4$. To overcome this, it is proposed that if $s$ is a state that is not visited then $p_s$ is approximated using a sensible prior or imputation. In the Results Section the overall cooperation rate is used. Another approach to overcoming this measurement error would be to measure strategies in a sufficiently noisy environment.

We can measure how close a strategy is to being zero determinant using standard linear algebraic approaches. Essentially we attempt to find $\mathbf{x} = (\alpha, \beta)$ such that:

$$C\mathbf{x} = \tilde{\mathbf{p}} \tag{18}$$

where $C$ corresponds to Eqs (7)–(9) and is given by:

$$C = \begin{bmatrix} R - P & R - P \\ S - P & T - P \\ T - P & S - P \\ 0 & 0 \end{bmatrix} \tag{19}$$

Note that in general, Eq (18) will not necessarily have a solution. From the Rouché-Capelli theorem if there is a solution it is unique since rank($C$) = 2 which is the dimension of the variable **x**. The best fitting $\mathbf{x}^*$ is defined by:

$$\mathbf{x}^* = \text{argmin}_{\mathbf{x} \in \mathbb{R}^2} \|C\mathbf{x} - \tilde{\mathbf{p}}\|_2^2 \tag{20}$$

Known results [21–23] yield $\mathbf{x}^*$, corresponding to the nearest extortionate strategy to the measured $\tilde{\mathbf{p}}$. It is in fact an orthogonal projection of $\tilde{\mathbf{p}}$ on to the plane defined by (13).

$$\mathbf{x}^* = (C^T C)^{-1} C^T \tilde{\mathbf{p}} \tag{21}$$

**Table 1. A seeded play of 20 turns of two strategies.**

| Turn | 1 | 2 | 3 | 4 | 5 | 6 | 7 | 8 | 9 | 10 | 11 | 12 | 13 | 14 | 15 | 16 | 17 | 18 | 19 | 20 |
|---|---|---|---|---|---|---|---|---|---|---|---|---|---|---|---|---|---|---|---|---|
| (8/9, 1/2, 1/3, 0) | C | C | D | D | D | C | D | D | D | D | D | C | C | C | D | D | D | C | D | D |
| Alternator | C | D | C | D | C | D | C | D | C | D | C | D | C | D | C | D | C | D | C | D |

The squared norm of the remaining error is referred to as sum of squared errors of prediction (SSE):

$$\text{SSE} = \left\| Cx^* - \tilde{p} \right\|_2^2 \tag{22}$$

This gives expressions for $\alpha, \beta$ as $\alpha = x_1^*$ and $\beta = x_2^*$ thus the conditions for a strategy to be acting extortionately becomes:

$$\frac{-x_2^*}{x_1^*} = \chi > 1 \ \text{ and } \ x_1^* \neq 0 \tag{23}$$

A further known result [21–23] gives an expression for SSE:

$$\text{SSE} = \tilde{\mathbf{p}}^T \tilde{\mathbf{p}} \tilde{\mathbf{p}} C (C^T C)^{-1} C^T \tilde{\mathbf{p}} \tag{24}$$

$$\text{SSE} = \tilde{\mathbf{p}}^T \tilde{\mathbf{p}} - \tilde{\mathbf{p}} C \mathbf{x}^* \tag{25}$$

Using this approach, the memory-one representation $\mathbf{p} \in \mathbb{R}^4$ of any strategy against any other can can be measured and if (23) holds then (24) can be used to identify if a strategy is acting extortionately. While the specific memory-one representation might not be one that acts extortionately or is even feasible (as noted in [4]), a high SSE does imply that a strategy is not extortionate. For a measured $\mathbf{p}$, SSE corresponds to the best fitting $\alpha, \beta$. Suspicion of extortion then corresponds to a threshold on SSE and a comparison of the measured $\chi = \frac{-\beta}{\alpha}$.

## Results

This section validates the approach of the previous section and present a number of numerical experiments to identify if strategies that perform strongly in evolutionary settings are close or not to the space of ZD strategies.

### Validation

To validate the method described, we use [19] which presents results from a tournament with 19 strategies with specific consideration given to ZD strategies. This tournament is reproduced here using the Axelrod-Python library [14]. To obtain a good measure of the corresponding transition rates for each strategy all matches have been run for 2000 turns and every match has been repeated 60 times. All of this interaction data is available at [17]. Note that in the interest of open scientific practice [17], also contains interaction data for noisy and probabilistic ending interactions which are not investigated here.

Fig 1 shows the SSE values for all the strategies in the tournament, as reported in [19] the extortionate strategy Extort-2 gains a large number of wins. Notice that the mean SSE for Extort-2 is approximately zero, while for the always cooperating strategy Cooperator the SSE is far from zero. It is also clear that ZD-GTFT2 defined as a ZD strategy does not act extortionately. This is evident by the fact that it does not rank highly according to wins which is due to its value of $\chi$ being less than 1. ZD-GTFT2 is referred to as a "generous" ZD strategy, other examples of this include ZDGen2 and ZDSet2 defined in [24]. The general performance of these will be discussed in the Results section.

Next, the results of a much larger tournament are presented. As a final validation of the proposed methodology here, Table 2 shows the theoretic values of $\chi$ versus the measured values for all ZD strategies in the tournament. The method accurately recovers $\chi$ from the observed play of the strategies. Furthermore, the SSE value is low for all of these. The values of SSE

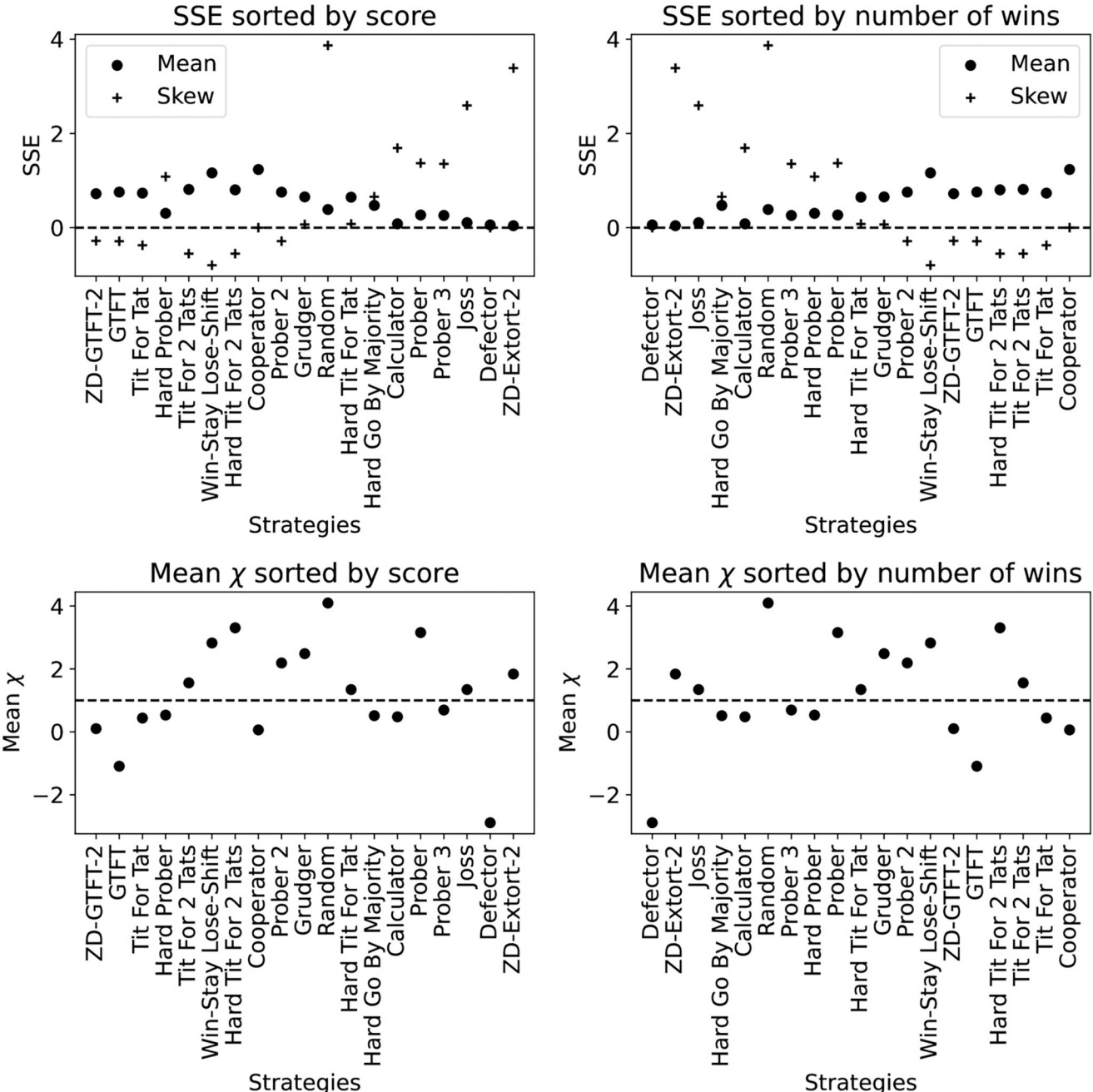

**Fig 1. SSE and best fitting $\chi$ for [19], ordered (in descending order) both by number of wins and overall score.** A win is when a strategy obtains a higher score than the player it is interacting with. The strategies with a positive skew SSE and high $\chi$ win the most matches, although even the theoretic extortionate strategy does not act in a perfectly extortionate manner in all matches. The strategies with a high score have a negatively skewed SSE.

above 1 indicate that whilst these strategies are designed to act extortionately they do not do so in all cases. This will be discussed in more detail in the next section.

## Numerical experiments

Next we investigate a tournament with 204 strategies. The results of this analysis are shown in Fig 2. The top ranking strategies by number of wins act in an extortionate way (but not against

**Table 2. Validating the approach by comparing the measured values of χ and the theoretic values of χ for all ZD strategies in the larger tournament.** The value of χ is effectively recovered from observed play and the SSE indicates that not all strategies are able to play as expected all the time.

| Name | Measured chi | Theoretic chi | SSE |
|---|---|---|---|
| Firm But Fair | 1.0000 | 1.0000 | 0.4446 |
| GTFT | 0.6999 | 0.7000 | 0.1373 |
| Joss | 1.2428 | 1.2431 | 0.0006 |
| Soft Joss | 0.9110 | 0.9112 | 0.0123 |
| Stochastic Cooperator | 3.0248 | 3.0276 | 0.2158 |
| Stochastic WSLS | 12.6105 | 12.6000 | 1.0627 |
| Win-Shift Lose-Stay | 1.8333 | 1.8333 | 1.4706 |
| Win-Stay Lose-Shift | 16.0000 | 16.0000 | 1.2353 |
| ZD-Extortion | 10.0067 | 10.0000 | 0.0000 |
| ZD-Extort-2 | 1.9978 | 2.0000 | 0.0000 |
| ZD-Extort3 | 3.0022 | 3.0000 | 0.0000 |
| ZD-Extort-2 v2 | 2.0020 | 2.0000 | 0.0000 |
| ZD-Extort-4 | 3.9998 | 4.0000 | 0.0000 |
| ZD-GTFT-2 | 0.8887 | 0.8889 | 0.0662 |
| ZD-GEN-2 | 0.8892 | 0.8889 | 0.0165 |
| ZD-SET-2 | 2.4022 | 2.4000 | 0.0661 |

all opponents) and it can be seen that a small subgroup of strategies achieve mutual defection. All the top ranking strategies according to score do not extort each other, however they **do** exhibit extortionate behaviour towards a number of the lower ranking strategies.

Note that while a strategy may attempt to act extortionately, not all opponents can be effectively extorted. For example, a strategy that always defects never receives a lower score than its opponent. As defined by [4], an extortionate ZD strategy will mutually defect with such an opponent which corresponds to the high values of $P(DD)$ seen in Fig 2 the top left quadrant.

A detailed look at selected strategies is given in Table 3. The high scoring strategies presented have a negatively skewed SSE whilst the ZD strategies have a low score but high probability of winning and higher probability of mutual defection. The skew of SSE of all strategies is shown in Fig 3 and supports the same conclusion. This evidences an idea proposed in [6]: sophisticated strategies are able to recognise their opponent and defend themselves against extortion. The high ranking strategies were in fact trained to maximise score [25] which seems to have created strategies able to extort weaker strategies whilst cooperating with stronger ones. Indeed unconditional extortion is self defeating.

## Evolutionary dynamics

In the original work introducing ZD strategies [4], effectiveness in evolutionary settings was already considered. Since then, most work surrounding these strategies considers their performance in evolutionary settings. Examples include [5–11]. The main motivation for this consideration is to gain insights on to how behaviours might arise but also whether or not they are stable in various settings such as social and biological interactions. Most of such work considers the space of memory-one strategies alone. In contrast, this paper considers a wider strategy space and two models of evolution are investigated: the continuous replicator dynamics and the discrete Moran process.

**Replicator dynamics.** From the large number of interactions a payoff matrix $S$ can be measured where $S_{ij}$ denotes the score (using standard values of $(R, S, T, P) = (3, 0, 5, 1)$) of the

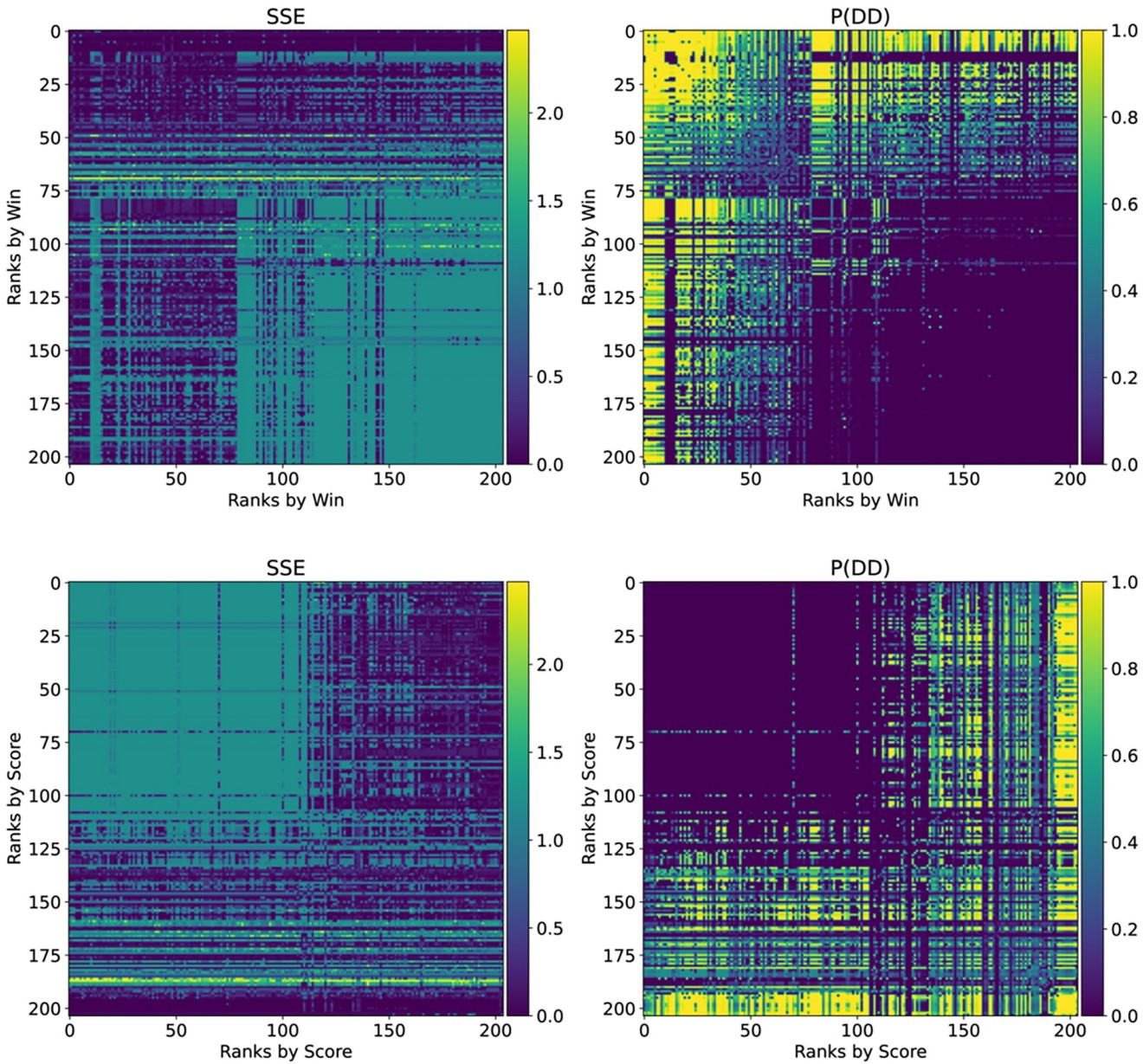

**Fig 2. SSE and probability of mutual defection ($P(DD)$) for the strategies for the full tournament.** The strategies with high number of wins have a low SSE however are often locked in mutual defection as evidenced by a high $P(DD)$. The strategies with a high score have a high SSE against the other high scoring strategies indicating that fixed linear relationship is being enforced. However against the low scoring strategies they have a lower SSE and against the very lowest scoring strategies a high $P(DD)$.

$i$th strategy against the $j$th strategy. Given a population of strategies represented by $\gamma$ where $\gamma_i$ denotes the proportion of the population occupied by the $i$th strategy, the fitness landscape under evolution can be considered. This is traditionally done using the replicator equation, describing the evolution of the population under selection:

$$\frac{d\gamma_i}{dt} = \gamma_i((S\gamma)_i - x^T S\gamma) \tag{26}$$

**Table 3. Summary of results for a selected list of strategies. Similarly to Fig 1, the high scoring strategies have a negatively skewed SSE.** The strategies with a large number of wins have a low SSE and positively skewed SSE. Note that a value of $\chi = 0.063$ and SSE = 1.235 corresponds to a vector $p = (1, 1, 1, 1)$ which highlights that the high scoring strategies, adapt and in fact cooperate often.

| Rank | Name | Score per turn | P(Win) | P(DD) | Median $\chi$ | Mean SSE | Skew SSE | Var SSE |
|---|---|---|---|---|---|---|---|---|
| 1 | EvolvedLookerUp2_2_2 | 2.944 | 0.230 | 0.092 | 0.062 | 1.057 | -0.857 | 0.160 |
| 2 | Evolved HMM 5 | 2.944 | 0.205 | 0.110 | 0.062 | 0.796 | -0.448 | 0.294 |
| 3 | PSO Gambler 2_2_2 | 2.913 | 0.204 | 0.128 | 0.062 | 0.899 | -0.508 | 0.255 |
| 4 | PSO Gambler Mem1 | 2.908 | 0.211 | 0.128 | 0.062 | 0.705 | -0.186 | 0.333 |
| 5 | PSO Gambler 1_1_1 | 2.906 | 0.221 | 0.145 | 0.062 | 0.737 | -0.209 | 0.296 |
| 7 | Evolved ANN 5 | 2.893 | 0.225 | 0.185 | 0.062 | 0.804 | -0.608 | 0.334 |
| 31 | ZD-GTFT-2 | 2.721 | 0.000 | 0.081 | 0.062 | 0.786 | -0.502 | 0.289 |
| 45 | ZD-GEN-2 | 2.689 | 0.016 | 0.096 | 0.062 | 0.694 | -0.227 | 0.358 |
| 69 | Tit For Tat | 2.638 | 0.000 | 0.157 | 0.062 | 0.773 | -0.507 | 0.301 |
| 75 | Grumpy | 2.630 | 0.075 | 0.100 | 0.062 | 0.978 | -1.438 | 0.245 |
| 88 | Win-Stay Lose-Shift | 2.616 | 0.099 | 0.122 | 0.062 | 1.172 | -4.501 | 0.027 |
| 103 | Eventual Cycle Hunter | 2.565 | 0.067 | 0.052 | 0.062 | 0.728 | -0.338 | 0.357 |
| 127 | Adaptive | 2.272 | 0.500 | 0.314 | -1.000 | 0.084 | 2.171 | 0.010 |
| 168 | ZD-SET-2 | 1.975 | 0.451 | 0.418 | 2.407 | 0.081 | 5.244 | 0.006 |
| 169 | Bully | 1.970 | 0.381 | 0.141 | -1.000 | 1.373 | -2.221 | 0.140 |
| 179 | Alternator | 1.945 | 0.392 | 0.259 | 3.857 | 1.332 | -1.021 | 0.120 |
| 181 | Negation | 1.941 | 0.356 | 0.141 | -1.000 | 1.470 | -3.204 | 0.083 |
| 182 | CollectiveStrategy | 1.931 | 0.915 | 0.762 | -2.888 | 0.085 | 6.082 | 0.028 |
| 183 | Cycler DC | 1.931 | 0.324 | 0.256 | 3.857 | 1.279 | -0.900 | 0.140 |
| 188 | Hopeless | 1.908 | 0.352 | 0.048 | 1.833 | 2.247 | -1.694 | 0.139 |
| 194 | Gradual Killer | 1.892 | 0.354 | 0.367 | 0.062 | 0.254 | 1.669 | 0.106 |
| 196 | Aggravater | 1.879 | 0.930 | 0.739 | -2.889 | 0.163 | 2.951 | 0.066 |
| 200 | ZD-Extort-2 | 1.821 | 0.851 | 0.652 | 2.005 | 0.019 | 5.435 | 0.009 |
| 201 | ZD-Extort-4 | 1.820 | 0.865 | 0.697 | 4.003 | 0.021 | 3.677 | 0.005 |
| 202 | ZD-Extort3 | 1.810 | 0.862 | 0.687 | 3.028 | 0.015 | 5.066 | 0.005 |
| 203 | Defector | 1.808 | 0.929 | 0.800 | -2.889 | 0.059 | 0.000 | 0.000 |
| 204 | Handshake | 1.806 | 0.870 | 0.737 | -2.888 | 0.126 | 3.825 | 0.083 |

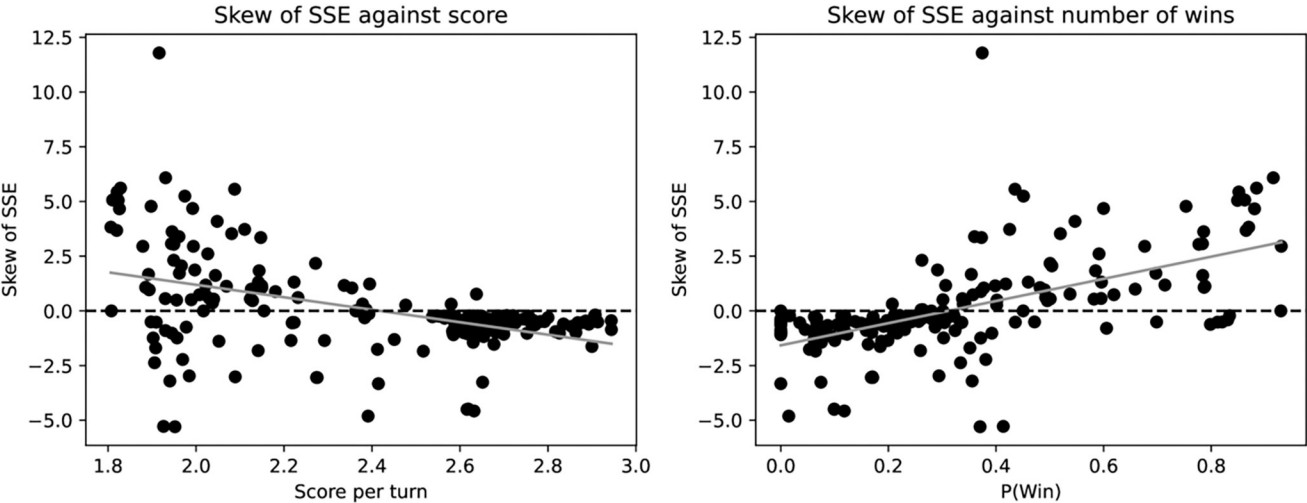

**Fig 3. Skew of SSE for all strategies considered over all opponents.** A similar conclusion to that of Fig 1 can be made: the strategies that score highly have a negatively skewed SSE highlighting their ability to adapt to their opponent. The auxiliary materials include a version of this graphic with strategy names.

Eq (26) is solved numerically for an initial population with a uniform distribution of the strategies. This is done using an integration technique described in [26] until a stationary vector $\gamma = s$ is found. Fig 4 shows the stationary probabilities for each strategy ranked by score. It is clear to see that only the high ranking strategies survive the evolutionary process (in fact, only 39 have a stationary probability value greater than $10^{-2}$).

Fig 5 plots the mean and skew (a standard statistical measure on a distribution) of SSE against the stationary probabilities **s** of (26). Strategies that perform strongly according to Eq (26) seem to be strategies that have a negative skew of SSE: indicating that they often have a high value of SSE (i.e. do not act extortionately) but have a long left tail allowing them to adapt when necessary. A general linear model obtained using recursive feature elimination is shown in Table 4 with stronger predictive power and confirming these conclusions.

Fig 6 shows the distribution of the SSE for three selected strategies. It is evident that Extort-2 almost always has the same low value of SSE against all opponents (which gives a positively

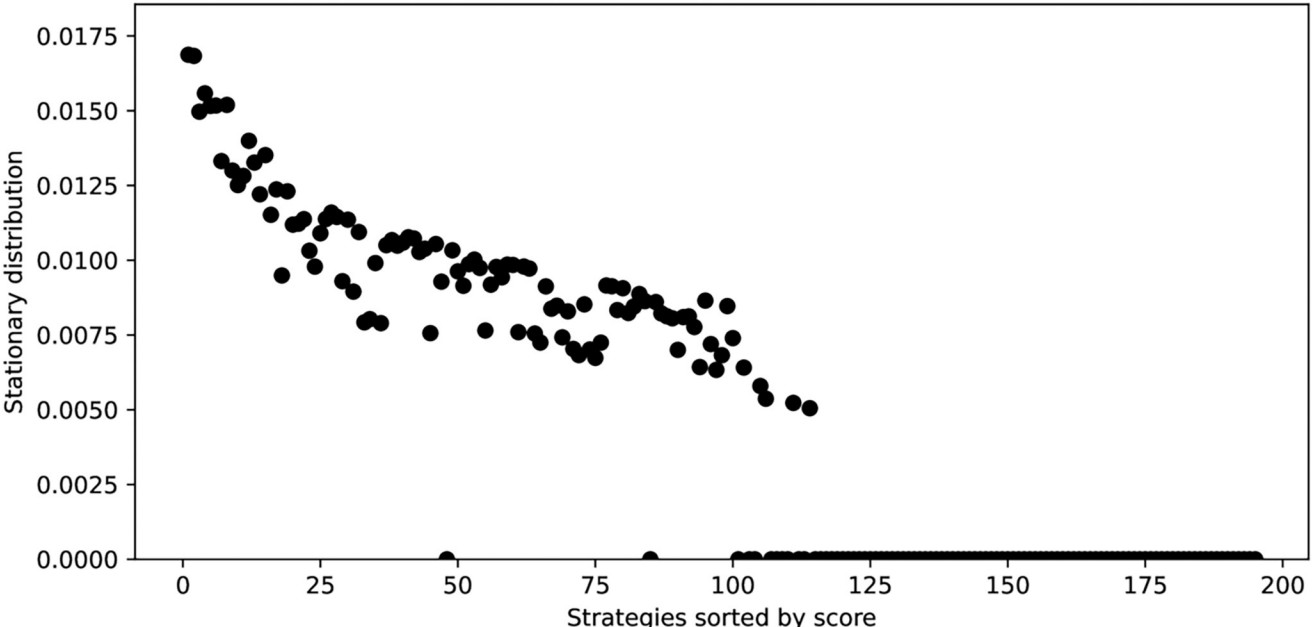

**Fig 4. Stationary distribution of the replicator dynamics (26): Strategies are ordered by score (as given in Table 3).** The 2 strategies with the highest stationary probability are: EvolvedLookerUp2_2_2 and Evolved HMM 5. Note that strategies that make use of the knowledge of the length of the game are removed from this analysis as they have an evolutionary advantage.

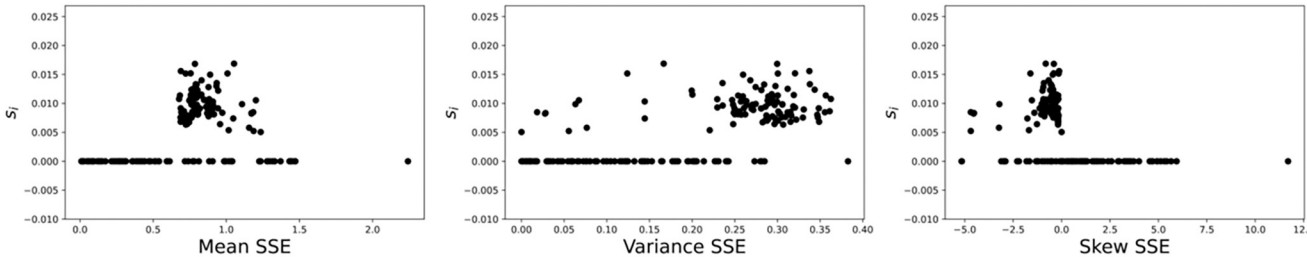

**Fig 5. Mean, variance and skew of SSE versus the stationary probabilities of (26).** The plot of the skew clearly shows that all high probabilities have a negative skew.

**Table 4. General linear model predicting the stationary probability as a function of the mean, median and variance of the SSE.** This shows that strategies with a low mean and high median are more likely to survive the evolutionary dynamics. This corresponds to negatively skewed distributions of SSE which again highlights the importance of adaptability.

| Dep. Variable: | | $s_i$ | | R-squared: | | 0.648 | |
| --- | --- | --- | --- | --- | --- | --- | --- |
| Model: | | OLS | | Adj. R-squared: | | 0.642 | |
| Method: | | Least Squares | | F-statistic: | | 117.0 | |
| | | | | Prob (F-statistic): | | 5.00e-43 | |
| | | | | Log-Likelihood: | | 851.41 | |
| No. Observations: | | 195 | | AIC: | | -1695. | |
| Df Residuals: | | 191 | | BIC: | | -1682. | |
| Df Model: | | 3 | | | | | |
| Covariance Type: | | nonrobust | | | | | |
| | **coef** | **std err** | **t** | **P > \|t\|** | **[0.025** | **0.975]** | |
| const | 0.0007 | 0.001 | 1.137 | 0.257 | -0.000 | 0.002 | |
| ('SSE', 'mean') | -0.0134 | 0.002 | -8.369 | 0.000 | -0.017 | -0.010 | |
| ('SSE', 'median') | 0.0139 | 0.001 | 10.433 | 0.000 | 0.011 | 0.017 | |
| ('SSE', 'var') | 0.0069 | 0.003 | 2.402 | 0.017 | 0.001 | 0.013 | |
| Omnibus: | | 17.190 | | Durbin-Watson: | | 1.664 | |
| Prob(Omnibus): | | 0.000 | | Jarque-Bera (JB): | | 25.453 | |
| Skew: | | 0.530 | | Prob(JB): | | 2.97e-06 | |
| Kurtosis: | | 4.418 | | Cond. No. | | 23.7 | |

skewed distribution), whereas EvolvedLookerUp2_2_2 and Tit For Tat have a wider distribution of values depending on the opponent (which gives a negatively skewed distribution).

**Finite population dynamics: Moran process.** The Moran Process is an evolutionary model of evolutionary in a finite population. Of specific interest is the probability of a single individual entrant to a population taking over the population. This is referred to as the fixation probability denoted by $\kappa_1$. In [11] a large data set of pairwise fixation probabilities in the Moran process is made available at [27]. Fig 7 shows linear models fitted to three summary measures of SSE and the mean (over population size $N$ and opponents) value of $\kappa_1 \cdot N$. This specific measure of fixation is chosen as $\kappa_1$ is usually compared to the neutral fixation probability of $1/N$. As was noted in [11], the specific case of $N = 2$ differs from all other population sizes which is why it is presented in isolation. We note that there is a significant relationship between the skew of SSE and the ability for a strategy to become fixed. A general linear model obtained through recursive feature elimination is shown in Table 5 which confirms the conclusions.

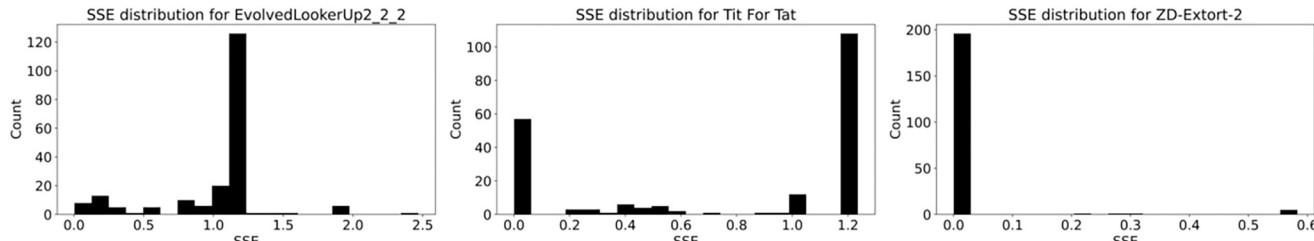

**Fig 6. Distribution of SSE values for 3 selected strategies.** The first two distributions are negatively skewed and the third has a positive skew.

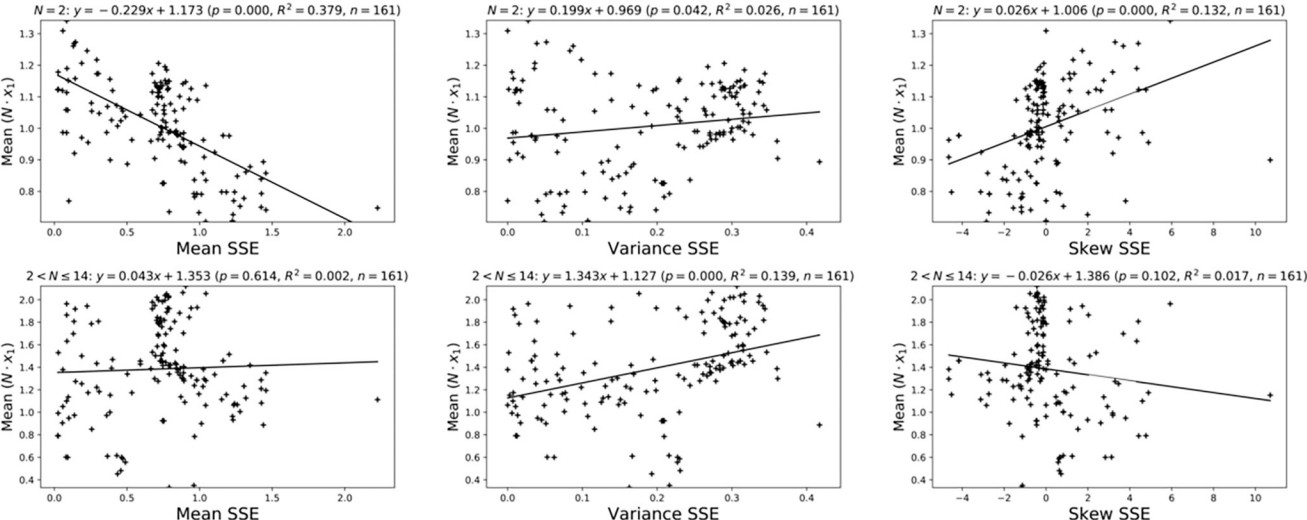

**Fig 7. The mean, variance and skew of SSE against the normalised pairwise fixation probabilities from [11] (for a given strategy averaged over all opponents and population sizes).** The clustering either side of a value of skew equal to 0 show that strategies with above neutra fixation ($N \cdot x_1 > 1$) negative skew.

These findings confirm the work of [11] in which sophisticated strategies resist evolutionary invasion of shorter memory strategies. This also confirms the work of [5, 6] which proved that ZD strategies where not evolutionarily stable due to the fact that they score poorly against themselves.

The work also provides strong evidence to the importance of adaptability: strategies that offer a variety of behaviours corresponding to a higher standard deviation of SSE are

**Table 5. General linear model predicting the mean fixation probability as a function of the mean, median and variance of the SSE.** This shows that strategies with a high mean and low median are likely to be evolutionarily stable. This corresponds to negatively skewed distributions of SSE which again highlights the importance of adaptability.

| Dep. Variable: | mean | | R-squared: | | 0.319 | |
|---|---|---|---|---|---|---|
| Model: | OLS | | Adj. R-squared: | | 0.310 | |
| Method: | Least Squares | | F-statistic: | | 36.53 | |
| | | | Prob (F-statistic): | | 9.74e-14 | |
| | | | Log-Likelihood: | | -42.272 | |
| No. Observations: | 159 | | AIC: | | 90.54 | |
| Df Residuals: | 156 | | BIC: | | 99.75 | |
| Df Model: | 2 | | | | | |
| Covariance Type: | nonrobust | | | | | |
| | **coef** | **std err** | **t** | **P > \|t\|** | **[0.025** | **0.975]** |
| const | 1.2815 | 0.056 | 22.993 | 0.000 | 1.171 | 1.392 |
| ('SSE', 'mean') | -1.0620 | 0.145 | -7.323 | 0.000 | -1.348 | -0.776 |
| ('SSE', 'median') | 0.9037 | 0.106 | 8.535 | 0.000 | 0.695 | 1.113 |
| Omnibus: | 2.302 | | Durbin-Watson: | | 1.716 | |
| Prob(Omnibus): | 0.316 | | Jarque-Bera (JB): | | 1.850 | |
| Skew: | -0.199 | | Prob(JB): | | 0.397 | |
| Kurtosis: | 3.348 | | Cond. No. | | 11.2 | |

significantly more likely to survive the evolutionary process. This corresponds to the following quote of [28]:

> *"It is not the most intellectual of the species that survives; it is not the strongest that survives; but the species that survives is the one that is able to adapt to and to adjust best to the changing environment in which it finds itself."*

## Discussion

This work defines an approach to measure whether or not a player is using an extortionate strategy as defined in [4], or a strategy that behaves similarly, broadening the definition of extortionate behavior. All extortionate strategies have been classified as lying on a triangular plane. This rigorous classification fails to be robust to small measurement error, thus a statistical approach is proposed approximating the solution of a linear system. This method was applied to a large number of pairwise interactions.

The work of [4], while showing that a clever approach to taking advantage of another memory-one strategy exists, is not the full story. Though the elegance of this result is very attractive, just as the simplicity of the victory of Tit For Tat in Axelrod's original tournaments was, it is incomplete and in the author's opinions, has been oversimplified and overgeneralized in subsequent work. Extortionate strategies achieve a high number of wins but they do generally not achieve a high score and fail to be evolutionarily stable.

Rather, more sophisticated strategies are able to adapt to a variety of opponents and act extortionately only against weaker strategies while cooperating with like-minded strategies that are not susceptible to extortion. This adaptability may be key to maintaining sustained cooperation, as some of these strategies emerged naturally from evolutionary processes trained to maximize payoff in IPD tournaments and fixation in population dynamics.

Following Axelrod's seminal work [1, 2], it was commonly thought that evolutionary cooperation required strategies that followed a simple set of rules. The discovery/definition of extortionate strategies [4] seemingly showed that complex strategies could be taken advantage of. In this manuscript it has been shown that not only is it possible to detect and prevent extortionate behaviour but that more complex strategies can be evolutionary stable. The complex strategies in question were obtained through reinforcement learning approaches [11, 25]. Thus, this demonstrates that it is possible to recognise extortion, both theoretically using SSE but also that this ability can develop through reinforcement learning. It seems human difficulty in directly developing effective complex strategies has been incorrectly generalized to a weakness in complex strategies themselves, which is demonstrable not the case. In fact, complex strategies can be the most effective against a diverse set of opponents.

A possible future research direction would be applying and or extending the methodology proposed here to consider other theoretic models of control of opponent utility such as [29–31]. There are however, various potential immediate applications for SSE, one of which could be to devise an agent that learns during the interactions with another agent. Fig 8 shows the average SSE value over a number of iterations over a number of repetitions. More investigation would be required but in some cases it seems that a large number of interactions would be required to gain certainly about the play of an agent. This approach seems to be in opposition of some of the trained strategies of [25] which are known to learn from early interactions and adapt their play.

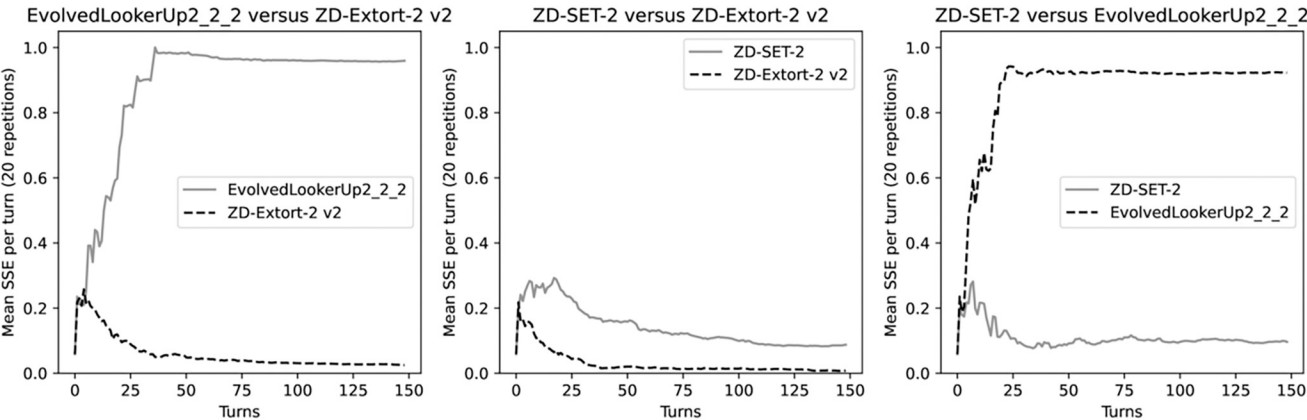

**Fig 8. The average SSE of a few strategies over a number of repetitions and a number of turns.** The EvolvedLookerUp2_2_2 strategy is recognisably not a ZD strategy after 10 turns against both opponents. When playing against the EvolvedLookerUp2_2_2 the generous ZD strategy ZDSet2 is also quickly recognisable as a ZD strategy after approximately 10 turns. Interestingly, while the other ZD strategy, ZD-Extort-2 v2, is clearly a acting as a ZD strategy early on against both opponents it would take longer to confidently recognise that that ZDSet2 is a ZD strategy.

In closing, the authors wish to emphasize the role of comprehensive simulations to temper theoretical results from overgeneralization, and perhaps more importantly, the ability of simulations to provide insights that are difficult to obtain from theory.

## Supporting information

**S1 Appendix. Proof of algebraic condition for extortionate strategies.**
(PDF)

**S2 Appendix. Skew of SSe for all strategies.** The skew for all the strategies in the larger tournament.
(PDF)

**S3 Appendix. List of all strategies.**
(PDF)

## Acknowledgments

The following open source software libraries were used in this research:

- The Axelrod [14, 15] library (IPD strategies and tournaments).

- The sympy library [32] (verification of all symbolic calculations).

- The matplotlib [33] library (visualisation).

- The pandas [34], dask [35] and NumPy [36] libraries (data manipulation).

- The SciPy [37] library (numerical integration of the replicator equation).

This work was performed using the computational facilities of the Advanced Research Computing @ Cardiff (ARCCA) Division, Cardiff University.

## Author Contributions

**Conceptualization:** Vincent Knight, Nikoleta E. Glynatsi.

**Data curation:** Vincent Knight.

**Formal analysis:** Vincent Knight, Marc Harper, Nikoleta E. Glynatsi.

**Investigation:** Nikoleta E. Glynatsi, Jonathan Gillard.

**Methodology:** Vincent Knight, Marc Harper, Nikoleta E. Glynatsi, Jonathan Gillard.

**Resources:** Nikoleta E. Glynatsi.

**Software:** Vincent Knight, Marc Harper, Nikoleta E. Glynatsi.

**Visualization:** Vincent Knight, Marc Harper, Nikoleta E. Glynatsi.

**Writing – original draft:** Vincent Knight, Marc Harper, Nikoleta E. Glynatsi.

**Writing – review & editing:** Vincent Knight, Marc Harper, Nikoleta E. Glynatsi, Jonathan Gillard.

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
