## [Decision Letter · Decision Letter 0]

29 Jan 2024

PONE-D-23-41274Recognising and evaluating the effectiveness of extortion in the Iterated Prisoner’s DilemmaPLOS ONE

Dear Dr. Knight,

Thank you for submitting your manuscript to PLOS ONE. After careful consideration, we feel that it has merit but does not fully meet PLOS ONE’s publication criteria as it currently stands. Therefore, we invite you to submit a revised version of the manuscript that addresses the points raised during the review process.

**ACADEMIC EDITOR: **The two reviewers have provided the constructive comments. They both agreed that the paper has good merit, and made several suggestions for further improvement. Please take them carefully into account when revising the paper.

We look forward to receiving your revised manuscript.

Kind regards,

The Anh Han, Ph.D.

Academic Editor

PLOS ONE

Additional Editor Comments:

The two reviewers have provided the constructive comments. They both agreed that the paper has good merit, and made several suggestions for further improvement. Please take them carefully into account when revising the paper.

Reviewers' comments:

Reviewer's Responses to Questions

**Comments to the Author**

1. Is the manuscript technically sound, and do the data support the conclusions?

Reviewer #1: Yes

Reviewer #2: Yes

2. Has the statistical analysis been performed appropriately and rigorously? 

Reviewer #1: Yes

Reviewer #2: Yes

3. Have the authors made all data underlying the findings in their manuscript fully available?

Reviewer #1: Yes

Reviewer #2: Yes

4. Is the manuscript presented in an intelligible fashion and written in standard English?

Reviewer #1: Yes

Reviewer #2: Yes

5. Review Comments to the Author

Reviewer #1: I reviewed this manuscript in another journal before and finally gave "Accept" in their revision 1. However, the handling editor gave "Reject" because he/she thought that the presentation of the paper is problematic.

I personally think this manuscript is worth publishing in PLOS ONE. However, at the same time, I think the points suggested by the handling editor at that time were also very reasonable. I just want to know if those points by the handling editor are clearly addressed in this submitted manuscript in PLOS ONE. If yes, please provide a detailed list of your revisions so I can check it.

Other points I'd like to mention. It is not natural to divide the abstract into two paragraphs. It should be one paragraph. Also, there are too many paragraphs in one section on average. Could you make it lesser overall?

Introduction: R, T, S, and P in the PD parameters should not be itemized. It is better to write them as a part of text.

Reviewer #2: In this work, the authors address the issue of establishing and maintaining mutual cooperation in in agent-to-agent interactions thanks to the direct reciprocity. Specifically, they consider the Iterated Prisoner's Dilemma and focus on the mechanism of extorsion, which represents a specific type of zero-determinant strategies. The idea behind the extortionate strategies is that agents cooperate at a small cost with the expectation of future benefits. Most works on zero-determinant strategies evaluate their effectiveness by placing them in specific scenarios and assessing their performance either numerically or theoretically. Moreover, extortionate strategies are defined as algebraically rigid and memory-one, requiring complete knowledge of a strategy's cooperation probabilities. Thus, the main contribution consists of proposing a method able to detect extortionate behavior from the play history of an arbitrary strategy. This approach differs from previous studies that observed the effectiveness of theoretically extortionate strategies. Rather, the study empirically quantifies the extortionateness of the largest known collection of strategies.

The proposed approach is validated empirically through a large tournament involving 204 strategies, showing that sophisticated strategies can recognize extortionate behavior and adapt to opponents. The study emphasizes the importance of adaptability for achieving evolutionary stability in the context of evolutionary dynamics.

Authors show how extortionate strategies achieve a high number of wins, but they do generally not achieve a high score and fail to be evolutionarily stable. Nevertheless, more sophisticated and complex strategies are able to adapt to a variety of opponents and act extortionately only against weaker strategies while cooperating with like-minded strategies that are not susceptible to extortion. This adaptability can be crucial to maintaining sustained cooperation, as some of these strategies emerged naturally from evolutionary processes trained to maximize payoff in IPD tournaments and fixation in population dynamics.

Overall, the work is well-structured and well-written and provides several interesting theoretical insights related to extortionate strategies. For instance, one behaviour observed from the experiments is that certain strategies can selectively behave extortionately toward some opponents while cooperating with others. Thus, while resistance to extortionate behavior is crucial for the evolution of cooperation, selectively extorting weaker opponents is possible and can be evolutionary advantageous.

In addition, the authors provide both the GitHub link with the code and all the data used are open-sourced and freely accessible.

In terms of presentation, both methods and results are rigorously presented and discussed. Please find below minor suggested corrections/improvement of the work:

1. Missing the reference to the section at the beginning of page 3: “In Section , the reverse problem is considered”

2. Equation (15) should be aligned to equations (13) and (14).

3. The second part of equation (24) can be a distinct equation, e.g. (25)

4. Although it is indicated in the title, Figs. 1 and 3 do not have an explicit label for the y-axis. I suggest including SSE as label of the y-axis.

5. A dot is missing in the first sentence of the subsection 3.3.2 before Figure 7. “In [24] a large data set of pairwise fixation probabilities in the Moran process is made available at [22]”

6. The quote reported at the end of section 3 could be in italics.

7. A comma is missing in the following sentence after “Rather”: “Rather more sophisticated strategies are able to adapt to a variety of opponents and act extortionately only against weaker strategies while cooperating with like-minded strategies that are not susceptible to extortion.”

6. PLOS authors have the option to publish the peer review history of their article (what does this mean?). If published, this will include your full peer review and any attached files.

Reviewer #1: No

Reviewer #2: No

---

## [Author Response · Author response to Decision Letter 0]

20 Mar 2024

# Response to reviewers

We would like to thank both reviewers for their comments on the manscript.

We attach a point by point response to each of their points, and in the case of

reviewer 1 a point by point response to each of the points raised by the editor

for a previous review.

We also include a marked up document showing the changes since our original submission to PLOS

One.

Although we not that this was done prior to changing the style of the manuscript

to match the style of PLOS One: this is to simplify how the marked up document

looks. Note that this marked up document does not include changes to any of the

mathematical equations.

## Reviewer 1

> I reviewed this manuscript in another journal before and finally gave "Accept" in their revision 1.

> However, the handling editor gave "Reject" because he/she thought that the presentation of the paper is problematic.

> I personally think this manuscript is worth publishing in PLOS ONE.

> However, at the same time, I think the points suggested by the handling editor at that time were also very reasonable.

> I just want to know if those points by the handling editor are clearly addressed in this submitted manuscript in PLOS ONE.

> If yes, please provide a detailed list of your revisions so I can check it.

We agree that the comments by the handling editor at the time were reasonable.

We took on their comments and made all the required changes.

The comments from the area editor were:

- It is not clear from Figure 1 whether sorting is made in an ascending or descending order from left to right. Readers have to guess it.

This has been clarified in the caption.

- Figure 1 and thereafter: the definition of "win" is not provided, and readers have to guess what it means.

A definition of a win has been added to the caption.

- The definition of P(DD) is not provided, so readers have difficulty in understanding Figure 2 and Table 3.

A definition of P(DD) has been added to the caption.

- Figure 4 presents a stationary distribution, but it is not clear how the initial

 condition of the replicator equation was chosen; in fact, because the replicator

 equation is 203=204-1 dimensional, there should be lots of equilibria in the system

 and the uniqueness of equilibrium is not at all guaranteed.

 In that case, the stationary distribution should be highly dependent on the

 initial condition, but the paper does not consider this problem and only

 cites "integration technique described in [29]".

The initial distribution chosen was a uniform distribution of the strategies.

This has been clarified in the text.

- It is not clear how many and what sorts of explanatory variables were used for

 general linear models with recursive feature elimination.

 Moreover, Tables 4 and 5 seem like just a copy-and-paste from a statistical software,

 and there are no descriptions about how to read/understand them.

 Readers have to guess everything, which is apparently a bad sign for a scientific paper.

 I find this is very critical, as Tables 4 and 5 are one of the main results.

 Furthermore, there are lots of pieces of information that are not relevant to the

 discussion in the main text

 (for example, "Date" and "Time" are obviously not necessary.

 Most readers could not understand item such as

 "Omnibus", "Durbin-Watson", "Jarque-Bera", "Cond. No." and so on,

 and they are not relevant to or even cited in the main text.)

The date and time have been removed and a few more details have been added to

the caption. We purposely included all other summary

measures of the linear regression process as good practice in research

communication. If the reviewer feels strongly to remove it we will do so but

would actually actively encourage all researchers to include as many measures

from processes like this as possible to ensure availability and reproducibility

of research.

- The fixation probability kappa_1 is discussed in Section 3.3.2

 (btw, x_1 in figure panels and in the legend remain uncorrected).

 However, there is no information about what type of Moran model

 was employed (e.g. how to transform payoffs to fitness,

 including the magnitude of selection strength), and only citations

 to [22] and [24] are given; readers have to look at these in order

 to figure out what assumptions are actually made.

 This is again problematic, because Figure 7 is one of the main results,

 based on which the authors claim that negative skew is a key to evolutionary success.

 Also, a justification is missing about why taking the average of

 (N kappa_1) over N=3 to 14 should be an appropriate way to measure evolutionary success.

More explanation has been given here.

- It is not clear why Eq.(23) is the condition for extortion.

 Eq.(23) means -beta < alpha, which means chi=(-beta)/alpha > 1 for alpha<0,

 but means chi=(-beta)/alpha < 1 for alpha>0.

We have corrected Eq.(23).

Another issue is that there are lots of grammatical unclarity throughout the manuscript, which prevents readers from smooth reading/understanding. To just list some from the first 3 pages;

- Abstract, "... lack thereof of zero-determinant strategies is done by placing some

 zero-determinant strategy": What is the meaning of "the lack of ZD is done by placing ZD"?

We have improved punctuation to hopefully clarify this sentence.

- Abstract: "their extortionateness quantified" -> "their extortionateness is quantified"

The wording has been fixed.

- Section 1, 3rd paragraph: "opponents previous play" -> "opponent's previous play"

- Section 1, 2nd last paragraph: "in to" -> "into"

- Section 2, 2nd paragraph: "by elements of R^4 mapping" -> "by elements of R^4, mapping"

- Below Eq.(6) "In Section, ": a section number is missing.

These have all been addressed.

- Below Eq.(15) "triangular (15) plane (13) in 3 dimensions (14)":

 This is not a reader-friendly way of citing equations.

We have modified this sentence.

- Below Eq.(15) "from [33] which is" -> "from [33], which is"

- Section 2.2, 2nd paragraph: "Table 1 which shows" -> "Table 1, which shows"

These have all been addressed.

> Other points I'd like to mention. It is not natural to divide the abstract into two paragraphs.

> It should be one paragraph. Also, there are too many paragraphs in one section on average. Could you make it lesser overall?

This seemed like a strange request from the original editor. If the reviewer

would like us to merge paragraphs we can do so. If not, we would prefer to keep

the presentation as is as we feel it improves readability.

## Reviewer 2

> 1. Missing the reference to the section at the beginning of page 3: “In Section , the reverse problem is considered”

This has been fixed.

> 2. Equation (15) should be aligned to equations (13) and (14).

The equality signs and strictly less than sign

are now all aligned.

> 3. The second part of equation (24) can be a distinct equation, e.g. (25)

This has been done.

> 4. Although it is indicated in the title, Figs. 1 and 3 do not have an explicit label for the y-axis. I suggest including SSE as label of the y-axis.

These have been added.

> 5. A dot is missing in the first sentence of the subsection 3.3.2 before Figure 7.

> “In [24] a large data set of pairwise fixation probabilities in the Moran process is made available at [22]”

This has been fixed.

> 6. The quote reported at the end of section 3 could be in italics.

This has been fixed.

> 7. A comma is missing in the following sentence after “Rather”: “Rather more sophisticated strategies are

> able to adapt to a variety of opponents and act extortionately only against weaker strategies while cooperating

> with like-minded strategies that are not susceptible to extortion.”

This has been fixed.

---

## [Decision Letter · Decision Letter 1]

16 May 2024

Recognising and evaluating the effectiveness of extortion in the Iterated Prisoner’s Dilemma

PONE-D-23-41274R1

Dear Dr. Knight,

We’re pleased to inform you that your manuscript has been judged scientifically suitable for publication and will be formally accepted for publication once it meets all outstanding technical requirements.

Kind regards,

The Anh Han, Ph.D.

Academic Editor

PLOS ONE

Additional Editor Comments (optional):

Reviewers' comments:

Reviewer's Responses to Questions

**Comments to the Author**

1. If the authors have adequately addressed your comments raised in a previous round of review and you feel that this manuscript is now acceptable for publication, you may indicate that here to bypass the “Comments to the Author” section, enter your conflict of interest statement in the “Confidential to Editor” section, and submit your "Accept" recommendation.

Reviewer #1: All comments have been addressed

Reviewer #2: All comments have been addressed

2. Is the manuscript technically sound, and do the data support the conclusions?

Reviewer #1: Yes

Reviewer #2: Yes

3. Has the statistical analysis been performed appropriately and rigorously? 

Reviewer #1: N/A

Reviewer #2: Yes

4. Have the authors made all data underlying the findings in their manuscript fully available?

Reviewer #1: Yes

Reviewer #2: Yes

5. Is the manuscript presented in an intelligible fashion and written in standard English?

Reviewer #1: Yes

Reviewer #2: Yes

6. Review Comments to the Author

Reviewer #1: (No Response)

Reviewer #2: The authors have addressed all of my minor comments and have further enhanced the manuscript's readability and presentation.

7. PLOS authors have the option to publish the peer review history of their article (what does this mean?). If published, this will include your full peer review and any attached files.

Reviewer #1: No

Reviewer #2: No

---

## [Editor Report · Acceptance letter]

20 May 2024

PONE-D-23-41274R1 

PLOS ONE

Dear Dr. Knight, 

I'm pleased to inform you that your manuscript has been deemed suitable for publication in PLOS ONE. Congratulations! Your manuscript is now being handed over to our production team.

Kind regards, 

on behalf of

Dr. The Anh Han 

Academic Editor

PLOS ONE